# Privacy-Preserving and Explainable AI in Industrial Applications

**Iulian Ogrezeanu \*, Anamaria Vizitiu, Costin Ciuşdel, Andrei Puiu, Simona Coman, Cristian Boldişor, Alina Itu, Robert Demeter, Florin Moldoveanu, Constantin Suciu and Lucian Itu**

Automation and Information Technology, "Transilvania" University of Braşov, 500036 Braşov, Romania; anamaria.vizitiu@unitbv.ro (A.V.); costin.ciusdel@unitbv.ro (C.C.); andrei.puiu@unitbv.ro (A.P.); simona.coman@unitbv.ro (S.C.); cristian.boldisor@unitbv.ro (C.B.); alina.itu@unitbv.ro (A.I.); rdemeter@unitbv.ro (R.D.); moldof@unitbv.ro (F.M.); suciuc@unitbv.ro (C.S.); lucian.itu@unitbv.ro (L.I.)
\* Correspondence: iulian.ogrezeanu@unitbv.ro

**Abstract:** The industrial environment has gone through the fourth revolution, also called "Industry 4.0", where the main aspect is digitalization. Each device employed in an industrial process is connected to a network called the industrial Internet of things (IIOT). With IIOT manufacturers being capable of tracking every device, it has become easier to prevent or quickly solve failures. Specifically, the large amount of available data has allowed the use of artificial intelligence (AI) algorithms to improve industrial applications in many ways (e.g., failure detection, process optimization, and abnormality detection). Although data are abundant, their access has raised problems due to privacy concerns of manufacturers. Censoring sensitive information is not a desired approach because it negatively impacts the AI performance. To increase trust, there is also the need to understand how AI algorithms make choices, i.e., to no longer regard them as black boxes. This paper focuses on recent advancements related to the challenges mentioned above, discusses the industrial impact of proposed solutions, and identifies challenges for future research. It also presents examples related to privacy-preserving and explainable AI solutions, and comments on the interaction between the identified challenges in the conclusions.

**Keywords:** artificial intelligence; industrial applications; privacy preservation; explainability; bias; fairness

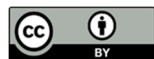

## 1. Introduction

Industry 4.0 [1] has introduced advanced technology in manufacturing, to make it more client-driven and customizable, leading to manufacturers striving toward a continuous improvement in quality and productivity. To achieve smart manufacturing, which enables variable product demand, intelligent systems were introduced in industrial units.

Recent developments in Internet of things (IOT) [2], Cyber-Physical Production Systems (CPPS) [3], and big data [4] led to major improvements in productivity, quality, and monitoring of industrial processes. Artificial intelligence (AI) plays an important role in industry, as more and more manufacturers are implementing AI in their processes.

Developed and employed with the purpose of performing tasks that normally require human discernment, AI is currently a popular topic. Having the capability of interpreting data for solving complex problems [5], AI is also a good fit for factories [6]. It enables industrial systems to process data, perceive their environment, and learn, while building up experience, in order to become better at a task by dealing with it and its data repeatedly.

Artificial intelligence [7] is a subject that researchers have been preoccupied with almost since computers were invented. AI includes every algorithm that enables machines to perform tasks that require discernment, not just by applying a formula or following a

strict rule-based logic. Thus, if we provide datasets with inputs and outputs to an AI algorithm, it will be capable of yielding a logic which maps the inputs to the outputs. In contrast, in classic programming, humans provide the logic. Of course, in many situations it is not necessary to use AI (e.g., if the problem can be solved through a mathematical formula). In the last two decades, thanks to the increase in computational power, AI has become very popular, and it has been used in several domains (medicine, marketing, industry, etc.), with various subdomains of algorithms such as machine learning (ML) [8] and deep learning (DL).

Machine learning is a popular subdomain of AI, and it is composed of statistical algorithms that can learn from data to create mathematical models for intelligent systems. Today, we have recommendation systems that use ML to suggest aspects that we like on the basis of our preferences (e.g., music, ads, and shopping). In the medical environment, we have ML models which help clinicians during diagnostic processes (decision support systems) [9].

Deep learning [10] is a widely used type of learning algorithm that relies on defining neural networks [11] with more than one hidden layer of neurons. Neural networks are inspired by the human brain, having computational units (named neurons) which are interconnected and exchange information to extract features from input data, thus enabling the mapping of input data to output data. However, neural networks used in AI do not work in the same way as human neural networks, because they exchange information using real numbers, whilst our neurons exchange information through electrical impulses. Neurons are organized in layers, where the first and the last layer are those that interact with the external environment, also named the input layer and output layer. Intermediate layers are called hidden layers.

Multiple review papers have described the multitude of approaches on the basis of which AI is employed in manufacturing. In [12], Sharma et al. presented a theoretical framework for machine learning in manufacturing, which guides researchers in elaborating a paper in this field. They pointed to several review papers that targeted the use of ML in the industrial environment. Rai et al. [13] discussed the use of AI in the context of the fourth industrial revolution. To highlight the potential advantages and potential flaws of using AI in industry, Bertolini et al. [14] reviewed the literature and classified research on the basis of the algorithm and application domain. Sarker [15] also reviewed the use of machine learning in real-world applications such as cyber security, agriculture, smart cities, and healthcare. In [16], Rao summarized the use of AI in different domains such as healthcare and travel.

In [17], four important challenges were identified: data availability, data quality, cybersecurity and privacy preservation, and interpretability/explainability. While the former two have been extensively discussed in the past and are well known, in this paper, we focus on topics related to the latter two challenges.

AI/ML relies extensively on existing and future data to deliver accurate and reliable results. The collection of large volumes of data for centralized processing poses severe privacy concerns. Thus, the first challenge refers to the fact that, while industrial data are abundant, they are hard to circulate and access due to privacy/IP constraints, also affecting the development of computer-based solutions. Industrial AI systems are difficult to realize, as data to develop and train them exist, but are not accessible. If training datasets lack diversity, algorithms may be biased or skewed to certain types of data/events [18].

Secondly, AI algorithms should be explainable and interpretable. ML algorithms are, in general, related to the concept of 'black box', i.e., the rationale for how the outputs are inferred from the input data is unclear [19]. Algorithmic decisions should, however, ideally provide a form of explainability [20]. In general, explanations are about the attribution of the worth of input features toward the final model predictions, whereas interpretability refers to the deterministic propagation of information from the input to the response function.

ML is usually regarded as a 'black box' unit; once a model is trained, its logic for determining the outputs on the basis of the inputs is not available, and further experiments and methods need to be performed to understand the way a trained model analyzes and processes the data. For stakeholders, however, it is important to understand how and why a solution is being proposed. Hence, explainable AI, with its interpretability tools, is key. Model-agnostic methods [21] were the subject of past research that yielded good results. Most of them targeted local interpretable model-agnostic explanations (LIMEs) [22] and Shapley additive explanations [23]. An important advantage of these methods is that they are compatible with a multitude of ML models. On the other hand, there are model-specific interpretation methods [24], which have the disadvantage of being compatible only with specific model types.

This paper highlights the recent developments related to privacy preservation and explainability in industrial AI applications and discusses the potential impact of existing solutions in the industrial domain. Several examples are presented, related to explainable AI methods and privacy preservation techniques. Section 2 addresses aspects related to privacy preservation in industrial AI applications, while the explainability and interpretability requirements of an AI model are discussed in Section 3. In the context of the approaches described herein, Section 4 focuses on the impact of AI in industry and identifies remaining challenges. Final conclusions are drawn in Section 5. Given the focus on the two challenges, the paper should be regarded as an argumentative review; the literature is examined selectively in order to support the arguments of the necessity of both explainability and privacy preservation in industrial AI applications. Furthermore, new challenges are identified, and their interaction is discussed.

## 2. Privacy Preservation in Industrial AI Applications

### 2.1. State of the Art in Privacy-Preserving AI

In this section, we briefly present various approaches for performing privacy-preserving AI.

One of the most used solutions in privacy-preserving AI is homomorphic encryption (HE). HE allows users to perform computations on encrypted data, yielding results that are also encrypted (results are identical to those obtained by performing the operations on unencrypted data). This type of encryption is necessary when processing sensitive data (e.g., healthcare data). Homomorphic encryption has been introduced and developed independently from AI, but the large computational overhead limits its real-world usage. Since AI-based methods provide results in near real time, i.e., the computational cost during inference is small, extending AI with HE allows for privacy-preserving data processing, while obtaining results in a reasonable amount of time.

One of the first notable approaches in using homomorphic encryption with neural networks was proposed by Orlandi et al. [25]. They developed an approach to process encrypted data using a neural network, ensuring that not only the data are protected, but also the neural network itself (weight values and hyperparameters). Figure 1 illustrates the data exchange between server and client, in an encrypted format.

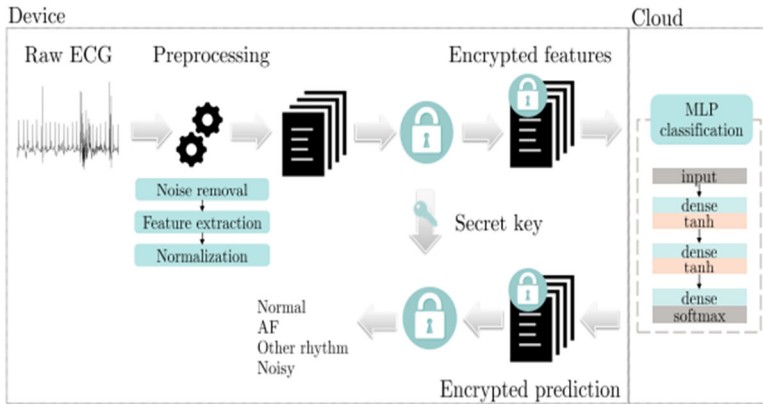

**Figure 1.** Workflow of privacy-preserving feature-based ECG classification method [26].

HE comes in many forms, and one of them is fully homomorphic encryption (FHE), which refers to a cryptosystem able to support arbitrary computation on ciphertexts. Sun et al. [27] used FHE to implement a private decision tree classification of user data. Aslett et al. [28] performed a review of homomorphic encryption techniques successfully applied in machine learning, and they also documented an R package implementing a homomorphic scheme. Deep neural networks were also employed in studies with homomorphic encryption. For example, Takabi et al. [29] used HE for multiparty machine learning; multiple parties participated in training the deep neural network, while maintaining data privacy.

A novel homomorphic encryption framework was proposed by Li et al. [30] to protect the data and the expertise of the algorithm when using cloud computing for model training. For healthcare and bioinformatics applications, Wood et al. [31] reviewed the use of FHE together with machine learning models.

The main disadvantage of FHE is its large computational cost. To address this aspect, partially homomorphic encryption (PHE) schemes were proposed and used by Fang et al. [32] to transmit encrypted gradients from all learning parties, thus speeding up the training by 25–28%, while maintaining the same level of accuracy in comparison with a classic approach.

Distributing the training process to multiple servers or decentralized edge devices with local data is a preferred approach to address privacy and scalability issues. This type of training is known as federated learning [33] and enables the collaboration of industrial nodes for training a model without exchanging sensitive data. However, reverse engineering may still be employed to extract from the model sensitive information regarding the datasets [34]. Hence, further research is needed for addressing privacy preservation.

Cloud-based implementations that can be used to run homomorphic encryption frameworks are available. One of them is Google's Cloud Platform [35], which runs on the same infrastructure as Google Search, YouTube, and Google Drive. Another widely used ML service is made available by Microsoft. Microsoft Azure Machine Learning [36] provides ML services which can also help in deploying models and managing them efficiently.

### 2.2. Review of Privacy-Preserving AI in Industrial Applications

In this section, we focus on research that targeted privacy-preserving artificial intelligence applied in industrial applications. Some of the main subjects in industry-oriented research are industrial Internet of things (IIOT) and Industry 4.0. A new method termed verifiable federated learning (VFL) was proposed by Fu et al. [37] for privacy preservation in industrial IOT, which employs federated learning, while also allowing for information extraction from the shared gradients. Figure 2 illustrates the proposed federated learning framework.

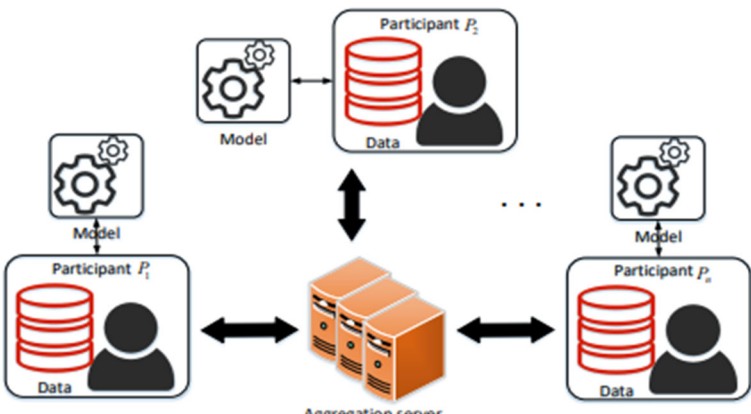

**Figure 2.** Federated learning framework proposed by Fu et al.

As mentioned above, a promising solution for privacy preservation is homomorphic encryption, but there are also other techniques for data encryption and/or anonymization. For example, on the basis of homomorphic data space transformation, Girka et al. [38] proposed an anonymization algorithm to protect data, while still allowing neural network training. They analyzed the effects that this method has on the neural network performance. By adding new frozen layers to the neural network, they succeeded in achieving anonymization, while the performance was slightly lower when compared to that of the original model.

Blockchain can also be used instead of simple federated learning. Zhao et al. [39] employed blockchain to transfer models trained by customers, thus eliminating the need for federated learning for gradient updates. Because blockchain records are not altered, malicious manufacturers or customer activities are traceable. Another study highlighted the need for privacy preservation to ensure data protection when exchanging information between multiple owners of renewable energy power plants. The main goal of the data exchange is to increase the forecast performance. Gonçalves et al. [40] proposed a privacy-preserving framework that combines the alternating direction method of multipliers with data transformation techniques. Their method proved to be successful, being robust to privacy breaches and communication failures, while the forecast performance was only marginally lower than that obtained using a model without privacy protection.

Generative adversarial network (GAN) is a machine learning algorithm that is widely used to generate synthetic data. Being first proposed in 2014 by Goodfellow et al. [41], GAN is currently popular and comes in different forms, one of them being least square generative adversarial network (LSGAN), which was used by Li et al. [42] together with federated learning to generate renewable scenarios. Through federated learning, a model was trained by gathering knowledge from different renewable sites, and then LSGAN was employed to generate renewable scenarios from the same distribution as the historical data, thanks to the capability of capturing the spatiotemporal characteristics of renewable powers.

Below, we provide a concrete example of a privacy-preserving AI application for casting [43]: a manufacturing process in which a liquid material is usually poured into a mold, which contains a hollow cavity of the desired shape, and then allowed to solidify. Defects may appear during the casting process, e.g., blow holes, pinholes, burr, shrinkage defects, mold material defects, pouring metal defects, and metallurgical defects, which have to be detected, and the corresponding parts have to be removed. Typically, this process is performed by a human operator, who may not be 100% accurate and consistent in their decisions. A fully automated, AI-based approach may reach 100% accuracy and remove inter- and intra-user variability, i.e., improve the robustness of the detection. The manufacturer would typically decide to externalize the development of the AI model,

which means that a large dataset containing photos of both acceptable and nonacceptable parts would have to be shared with the entity developing the AI model. However, the manufacturer may not feel comfortable with externalizing photos of nonacceptable parts. In a privacy-preserving setting, the photos would first be homomorphically encrypted or obfuscated, such that the external party cannot reconstruct the original images. The AI model would be trained on the encrypted or obfuscated images, and the trained model would be deployed as an AI service. During inference, the same encryption or obfuscation method would be employed to ensure that the AI model is not fed with out-of-distribution data. A possible technical solution was recently published for a healthcare application [44], which could be similarly applied in the industrial domain for the casting application. Therein, an image obfuscation algorithm was proposed that combines a variational auto-encoder with random non-bijective pixel intensity mapping to protect the content of medical images, which are subsequently employed in the development of DL-based solutions. Although a drop in accuracy could be observed when the classifier was trained on obfuscated images, the performance was deemed satisfactory in the context of a privacy–accuracy tradeoff.

## 3. Explainable Industrial AI Applications

### 3.1. The Black-Box Aspect of AI

Artificial intelligence algorithms are, in general, regarded as black-box algorithms, i.e., it is not possible to determine or infer why the model has generated a certain output. A representative example was described in [45]; a robotics graduate student tried in 1991 to train a military vehicle to self-drive. The training of the system was accomplished by him manually driving the vehicle while the system (the algorithm was a neural network) memorized the moves for different situations. After a few training sessions, the approach seemed to be working well; however, when the vehicle reached a bridge, it did not know how to handle the situation, and the model would have crashed the vehicle if the user had not intervened. Further testing revealed that the model was relying on grassy roadsides to be guided along the road; hence, the appearance of the bridge caused confusion.

The black-box problem [46] has represented a concern since the very beginnings of neural network research. Currently, very complex neural network architectures are employed, which deepen the black-box problem. The advancements of the technology and of computing power have also further increased its importance. It has become obvious both as a developer and as a user that, to trust an algorithm for making important decisions, one needs to make sure that the algorithm relies on the right properties and reasons.

### 3.2. State of the Art in Explainable AI

To make it easier for a user to understand the logic behind the decision taken by an algorithm, a user interface (UI) should accompany it. Using UI, automation bias can be mitigated. We can categorize explainable AI methods into the following:

- saliency maps: elements in the input that have the largest influence in the prediction are identified (e.g., LIME);
- feature attribution: attributing the classification to a small number of numeric/semantic features [47,48];
- metric learning [49]: mapping out data structures by deriving a metric from a classifier (explicit Siamese networks are very popular);
- activation maximization: methods that are based on GAN.

There are several methods of explainable AI that can be applied, depending on the data type. These are discussed below.

Even though there are several explainable AI techniques, only certain methods can be applied on tabular data [50]. Techniques designed for images or text data are typically not applicable to tabular data. Tabular data can present characteristics such as correlations between features or temporal aspects, and they may contain categorical features along

with continuous features. Poulin et al. [51] proposed one of the first methods of explainable AI, named ExplainD. It measures the importance of each input feature related to the prediction of a classifier. LIME proposed by Ribeiro et al. [52] is a model-agnostic technique used to explain the predictions of the classifier. Many other methods have been developed on the basis of LIME. The core principle is that one or more models are trained to approximate the predictions of the classification model, to determine why the classification model outputted a certain prediction. These models are trained with a dataset containing perturbed data points, which are close to the instance of interest. The newly trained model (named the surrogate model) is used to compute the proximity of each sample instance to the instance of interest. In Figure 3, the explanations for three predictions can be observed.

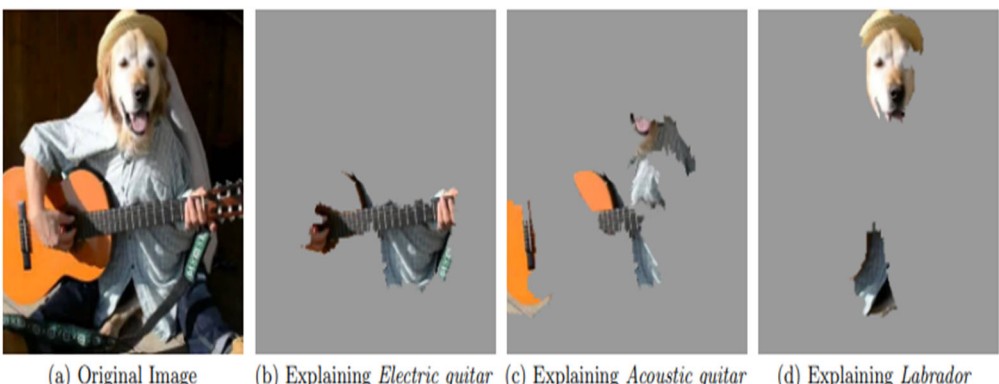

(a) Original Image    (b) Explaining *Electric guitar*    (c) Explaining *Acoustic guitar*    (d) Explaining *Labrador*

**Figure 3.** Explaining Google's Inception neural network predictions: electric guitar (**b**), acoustic guitar (**c**) and Labrador (**d**). Highlighted parts from the (**a**) original image are those that contributed the most to each prediction. Adapted with permission from [52], 2016.

For time series, saliency maps can be extracted to highlight the importance of a sequence from the input, which is related to the prediction. Class activation mapping (CAM) is a method used on convolutional neural networks (CNN) to identify input features which are representative for a class. Oviedo et al. [53] applied this method to explain model decisions when classifying small X-ray diffraction. Thrun et al. [54] proposed a new method of explainable artificial intelligence (XAI) in which a data-driven approach is used to exploit distance-based data structures, without the need for making any assumption related to the data. LIME can also be employed, as well as any method that yields a heatmap, e.g., deep learning important features (DeepLIFT). Layer-wise relevance propagation (LRP) [55] can also be used, which is a method that computes feature importance by backpropagating a relevance score through the model. Figure 4 contains an example of how features are highlighted to justify model predictions.

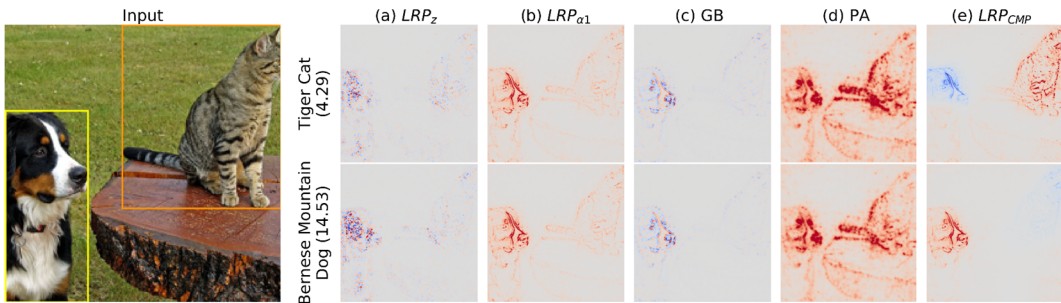

**Figure 4.** Heatmaps yielded by different techniques for two animals identified in a picture. Adapted with permission from [55], 2020.

It is widely known that XAI presents a wide variety of methods to be used for imaging data. All methods described in the categories above can be applied to gain insight into

the logic used by the model to perform the prediction. Yang et al. [56] discussed the use of XAI for medical images, and Xu et al. [57] described approaches and future challenges after summarizing the history of XAI.

### 3.3. Review of Explainable Industrial AI Applications

Manufacturers are now more interested in the use of AI to improve the overall quality of industrial applications, while at the same time unboxing the original black-box models. One of the domains in which AI is not widely used is air traffic management (ATM); however, introducing AI is a need which can help ATM in the future. As they are increasing in complexity, there is also a need for explainable AI techniques, to identify the most important features in model predictions [58]. Gade et al. [59] introduced a tutorial in which they presented an overview of model explainability and interpretability, alongside techniques which are helpful in providing explainability for AI systems. Their examples were mainly applications from the industrial environment. Longo et al. [60] and Ahmed et al. [61] also reviewed XAI applied in industry. In the automotive field, explainable AI is used to create transparency and understand model decisions [62], but it turns out that XAI itself is not sufficient to increase the trust [63]. Other explanations need to be provided, not only for developers, but also for the end-user.

Krishnamurthy et al. [64] proposed a new XAI framework for predicting maintenance applications for automotive applications and others. Brito et al. [65] used XAI for diagnosis and fault detection in rotating machinery. Anomaly detection was used for performing, while, for interpreting fault diagnosis models, they employed Shapley additive explanations (SHAPs). To reduce energy consumption in mineral processing, Chelgani et al. [66] highlighted the importance of using high-pressure grinding rolls (HPGRs). The main problem is HPGR modeling; they proposed to expand the existing conscious laboratory (CL) and used XAI systems to innovate powder technology industries. SHAP and extreme gradient boosting (XGBoost) were the selected methods for achieving model explainability.

## 4. Industrial Impact and Remaining Challenges

### 4.1. Industrial Impact

The use of artificial intelligence in industrial applications, and the use of techniques for privacy preservation and model explainability can impact industries in many ways:

- improving productivity: by predicting the quality parameters of the product [67], manufacturers can swiftly modify the industrial process setup to fit the updated requirements. Thus, they can save time by using an AI method to provide the best setup which meets their needs;
- improving maintenance: AI algorithms can be used to identify anomalies, and they can also handle large quantities of data [68]. By training an AI model to behave like a device in its normal state, it will be able to identify events that are abnormal (anomalies), which are dangerous, and which can lead to accidents. Prevention is a key factor in reducing them;
- increasing security: the use of privacy-preserving methods for artificial intelligence algorithms will increase security for the client, as well as the provider, making sure that no entity can have access to the model expertise while using the model to generate predictions [69];
- increasing trust in predictions: to improve the accuracy, the model complexity must be increased, and this leads to models being regarded as black boxes. To identify the logic behind a prediction, explainability methods have been developed, and they can be used to identify key features from the input data, which lead to a certain prediction and, thus, an understanding of the model logic.

## 4.2. Remaining Challenges

In this section, we present and discuss other remaining challenges related to the use of AI in industrial applications. AI models come with inherent risks posed by factors which are outside our control or governance, such as biased datasets or lack of robustness. These are discussed briefly below.

### 4.2.1. Bias and Fairness

Data remain the base of every learning algorithm, and any issue in the underlying dataset will be reflected in the algorithm performance. One such example is data bias [70]. If datasets present biases, then these will be learned and reflected by the model predictions. Biases can also appear when data are not biased, e.g., due to design choices. Considering the training of a neural network for identifying animals, like dogs, for example, if 95% of dogs in the dataset have brown fur, then the model will not have a good performance on dogs with white fur, and the model might focus on color instead of key features to identify dogs in images.

In the industrial environment, it is crucial to store data yielded by each hardware component (sensors, motors, etc.), so that an algorithm can extract relevant features and learn to predict accurately. Typically, equipment is running in a normal state, and breakdowns are very rare, e.g., once a week, month, or even year. Hence, omitting breakdown events will result in a biased dataset, because the model will react poorly when an abnormal event happens, thus decreasing the trust in its predictions.

Another issue that can be encountered in datasets related to industrial applications is data noise [71]. Training only on noisy data may result in a biased model because, when presenting data without noise or with a decreased level of noise, the model may have a weaker performance. Trying to filter the noise from a dataset may not be a good choice either, since noise happens uncontrollably and is typically present in real-world applications. To address these issues, one needs to make sure that datasets include both normal and noisy data, representative for the actual use of the models.

Another overarching topic is that of fairness [72]. There is no general definition since it is a term which spans across multiple domains such as computer science, psychology, and philosophy. A fair algorithm will be one that is not biased and does not discriminate against individuals, groups, or subgroups. If biases are identified in datasets, it is necessary to eliminate them before performing the training, to ensure that the model will not perpetuate them.

### 4.2.2. Robustness

Robustness refers to the property that characterizes how effective the AI model is when being tested on a new independent dataset. Specifically, robustness can be linked to the topic of confidence and out-of-distribution detection. It is known that the output of classic deep neural networks may be unreliable when applied on out-of-domain, noisy, or uncertain input data. Many methods have been proposed for assessing model output confidence.

Normalizing flows (NFs) are a family of generative models with tractable distributions, where both sampling and density evaluation can be efficient and exact [73]. The goal is to model $p(x)$, where $x$ denotes samples from a training set and $p(x)$ is the probability distribution. An NF model can answer the following question: given a new set of $x$, how likely are they to be from the same $p(x)$ distribution (as observed in the training set)? The NF framework employs two components: a bijective encoder (usually employing deep neural networks) and a prior probability distribution (usually a fixed multivariate normal distribution). In contrast with other methods such as variational inference (which only offer a lower bound of $p(x)$ named "evidence lower bound-ELBO"), NFs are capable of fast density estimation, given a suitable choice of model architecture. In that aspect, coupling layers have recently been proposed [74,75] which offer a simple and efficient

mechanism for computation of both forward (for density estimation) and backward passes (for sampling). Training can be performed end-to-end in an unsupervised manner.

An NF model can be deployed to detect out-of-distribution (OoD) input samples which should be excluded from the downstream deep neural network (DNN) pipeline. For example, given a model which was trained on a supervised task on a trainset T, an NF model can be trained on the same trainset. If, for new samples, the NF model computes low probability estimates, then those samples are outside the training manifold T of the supervised model, and its predictions may be regarded as unreliable.

Another approach to OoD detection in multiclass classification tasks is to employ softmax logits to compute energy scores, which have been shown to be aligned with the probability density of the inputs and be less susceptible to the overconfidence issue [76]. This approach can distinguish between in- and out-of-distribution samples, even when employing out-of-the-box models which have not been specifically tuned for this purpose. Naturally, OoD detection can be improved by employing outliers and an additional loss term which encourages the network to maximize the separation between energy scores for true samples vs. outliers. They have also stated that "methods relying on the softmax confidence score suffer from overconfident posterior distributions for OoD data". This means that output softmax probabilities tend to be erroneously high for outliers. It is shown that a classification model having a softmax final activation implicitly contains an input density estimator. The energy score can also be incorporated into the training objective, along with the categorical cross-entropy classification loss. The training dataset would consist of two parts: (i) an in-distribution subset, on which the classification loss is computed, and for which the energy score is minimized, and (ii) an OoD subset, on which only the energy score is computed and maximized. The energy method is applicable to an already trained model, extending it as an OoD classifier. Moreover, fine-tuning energy scores during main training can boost OoD detection performance.

Other approaches for analyzing and improving robustness is to use uncertainty estimation techniques. An established method for uncertainty quantification is to employ Gaussian processes (GPs). Figure 5 illustrates the performance of two methods using GPs for uncertainty quantification: red dots are the predictions, deep blue lines represent the label values, and light blue areas represent the certainty limits. In their original formulation, they lack efficiency for large dimensional datasets, but recent studies showcased that using variational approaches and certain architectural constraints, highly efficient models can be obtained, which offer high task-specific performance and uncertainty estimates using only a single forward pass of the neural network [77]. A smooth and sensitive feature extractor feeds a sparse variational Gaussian process, which outputs both the expected prediction and the output variance.

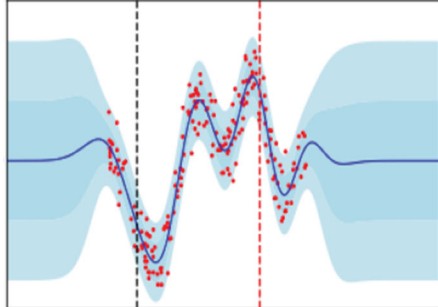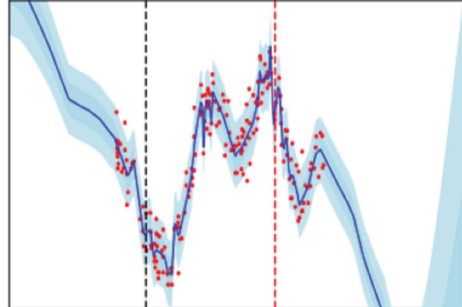

**Figure 5.** Predictions and attached uncertainties on a toy dataset from a squared exponential (SE) kernel (**left**) and a deep kernel learning (DKL) kernel (**right**). Outside the training points, the SE kernel model reverts to its zero mean with high uncertainty, while the DKL extrapolates confidently (adapted with permission from [77], 2021).

To improve robustness for deep neural network models, Kwon [78] proposed the concept of a backdoor attack, to retrain the model with additional altered images that were

intentionally introduced to mislead the algorithm. A backdoor attack relies on introducing inputs that contain certain triggers which determine the model to output a wrong prediction. After model training, triggers may be introduced in the test input data to try to manipulate the predictions. The advantage of the proposed method is that the classification accuracy is maintained at a high level, even when the model is under attack.

Text datasets are also subject to potential robustness issues. These may be addressed similarly to imaging datasets, i.e., by adding noise over some of the samples used in the training process. These samples with noise are also known as adversarial examples; noise is represented by modified characters or words in a paragraph. As for images with noise, text altered by noise should have the same meaning, i.e., a human reader should perceive it similarly to text without noise. By introducing adversarial examples in the training dataset, Kwon and Lee [79] increased the model accuracy over altered samples by 13.3% (from 9.2% to 22.5%), while the overall accuracy dropped only by 0.9% (from 88.1% to 87.2%).

## 5. Conclusions

Artificial intelligence is widely used and brings benefits in every domain in which it is applied. Industry 4.0 has created the conditions to apply artificial intelligence through digitalization. Large quantities of data are now available to be used to generate knowledge, which is exactly what AI algorithms have been designed for.

Even though it can improve industrial processes, AI needs to be applied with caution, because the access to large quantities of data also has downsides, such as the risk of data theft. To prevent this, privacy-preserving methods have been developed to be used along with AI algorithms, to ensure that knowledge is generated, while maintaining data safety. Privacy-preserving solutions proposed to date have the disadvantage of either increasing the runtime by a prohibitive amount or decreasing the accuracy significantly. Thus, the tradeoff between privacy preservation and usability is still too large. Further research is warranted to develop solutions which can be considered both secure and accurate enough.

Another aspect related to artificial intelligence is the black-box nature of the models. Increasing the accuracy means increasing the model complexity, thus making it harder to interpret how a model takes the decision starting from the given inputs. Explainable AI has been introduced to fill this gap and to help understand the way a model maps inputs to outputs. Current solutions are limited to certain models, and state-of-the-art AI approaches, such as deep neural networks, require further research to ensure levels of transparency that would allow the user to fully trust AI model decisions.

As AI techniques evolve, newly developed concepts will be translated into the various application domains, including industry. Similarly, new challenges will be identified, which will need to be addressed first at a core or theoretical level, and then within the application domains. Two such current challenges were described herein: bias/fairness and robustness. AI model robustness is closely linked to AI model explainability:;; a robust model will perform well even when being presented with a data sample that has distinct properties from those of the training data samples. A robust AI model performs well on such out-of-distribution data, specifically because it is capable of recognizing and interpreting certain characteristics of the data sample, even if they have a slightly different appearance than in the training dataset. Such model capability is also crucial for achieving high model explainability; a model that generalizes well takes the decisions on the basis of the right characteristics of the data samples, which in turn means that it can potentially explain its decisions correctly, i.e., generating trust. Furthermore, AI model robustness is also linked to AI model bias. A model without or with low bias is likely to achieve a superior robustness by removing or at least reducing so-called 'blind spots' in the data processing and interpretation. We also note that privacy-preserving methods increase complexity, since they introduce an additional layer of data manipulation. Specifically, the data manipulation methodology itself should not introduce any bias in the data and



maintain the same level of robustness as if the model was trained on the original data. Lastly, we note that privacy preservation and explainability requirements apparently have opposite effects. Privacy preservation is typically achieved by encrypting/obfuscating/altering the model input data, which in turn diminishes the explainability and interpretability capabilities. Hence, at least with current approaches, the user has to choose which of these aspects should be prioritized.

As a limitation, we note that this argumentative review does not represent an exhaustive attempt at discussing the application of AI in industrial applications. We focused on specific challenges and highlighted recent developments related to these challenges, and we identified other challenges to be considered in future research.

**Author Contributions:** Conceptualization, I.O. and L.I.; methodology, I.O., A.V., C.C., and A.P.; validation, S.C., C.B., A.I., R.D., F.M., and C.S.; resources, L.I.; writing—original draft preparation, I.O., C.C., A.P., and L.I.; writing—review and editing, R.D., F.M., and C.S.; visualization, I.O.; supervision, C.S.; project administration, L.I.; funding acquisition, L.I. All authors read and agreed to the published version of the manuscript.

**Funding:** This work was supported by a grant from the Romanian National Authority for Scientific Research and Innovation, CCCDI–UEFISCDI, project number ERANET-PERMED-PROGRESS, within PNCDI III.

**Institutional Review Board Statement:** Not applicable.

**Informed Consent Statement:** Not applicable.

**Data Availability Statement:** Not alicable.

**Conflicts of Interest:** The authors declare no conflict of interest.

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
