# Peer review of "Privacy-Preserving and Explainable AI in Industrial Applications"

_applsci, doi:10.3390/app12136395_

Round 1
Reviewer 1 Report
Comment:
I would like to commend the authors for putting this manuscript together. The authors have presented a review on the use of artificial intelligence techniques in industrial applications and the need to address the challenges of low industrial data due to the use of the encryption schemes by industries. The authors have made some attempts to address the main question posed. However, the following issues must be addressed to improve the quality of this manuscript:
(1) There is a need for the authors to clearly underline the contributions of this manuscript.
(2) I recommend that the authors should compare their work with the relevant studies.
(3) The authors should logically describe the relationship between the use of artificial intelligence and homomorphic encryption for privacy-preservation in industrial applications in section 2.
(4) The authors should clearly include the limitations of the study in the conclusion section.
Reviewer 2 Report
This paper deals with an exciting topic. The article has been read carefully, and some minor issues have been highlighted in order to be considered by the author(s).
#1 What is the motivation of this paper?
#2 What is the contribution and novelty of this paper?
#3 What is the advantage of this survey paper?
#4 Which evaluation metrics did you used for comparison?
#5 It would be good if security domains for the deep neural network would be reflected in the related work such as Medicalguard: U-net model robust against adversarially perturbed images, BlindNet backdoor: Attack on deep neural network using blind watermark, Textual Adversarial Training of Machine Learning Model for Resistance to Adversarial Examples, Ensemble transfer attack targeting text classification systems.
#6 In Figure 4, it is required to improve the resolution of the figures.
#7 In Figure 5, what is a blue line, a red point, and a green area?
Reviewer 3 Report
In this paper, the authors focus on recent advancements related to the challenges of privacy preservation while ensuring acceptable performance for AI applications. They discuss the industrial impact of proposed solutions, and identify challenges for future research. They also present few examples related to privacy-preservation and explainable AI solutions. Authors address few aspects related to privacy preservation in industrial AI applications, while considering the explainability and interpretability requirements of an AI model. The impact of AI in industry is briefly mentioned.
Authors claim that the paper may be “regarded as an argumentative review: literature is examined selectively in order to support the arguments of the necessity of both explainability and privacy preservation in industrial AI applications. Furthermore, new challenges are identified and their interaction is discussed.”
Though the topic is research is very relevant and significant, there are major issues with the paper that should be addressed.
1. Lack of systematic literature review: There is no clearly defined problem statement. It seems that authors have tried to put multiple ideas together in a document that lacks structure and coherence. There is no basis provided for inclusion or exclusion of documents and/or sources.
2. No critical review: There is lack of critical discussion on any research question. It seems to be a mix of excerpts from various papers.
3. No novelty: The challenges identified are common to all AI/data mining applications and not unique to the IIoT scenario. Problems of privacy preservation, privacy utility trade-off, noise in data, bias, etc are well known challenges.
4. No analysis of the literature has been presented. Authors could have proposed some taxonomy of methods for privacy preservation in IIoT AI applications, or proposed some techniques of adding interpretability to models, etc.
Overall, the paper lacks scientific vigour and novelty to be considered for publication.
Reviewer 4 Report
The submitted manuscript is a review paper on privacy-preserving and explainable artificial intelligence in industrial applications. The article is, in general, nice to read. The basic theory is also well explained.
What I am missing the most, is any information about how was the review performed. Which keywords and scientific databases have been used to pick up the relevant papers? Which criteria (inclusion and exclusion) have been used to include a paper? Answers to these questions should definitely be added to the submitted manuscript, in order to make it clear and more objective.
Some minor comments include:
A shortcut AI should be added to the abstract in the first occurrence of artificial intelligence (line 5 of the abstract).
Item points on page 2 (sentence starting with “Within [17] four important challenges …”) should rather be written in one line, separated by a colon or semicolon. Now they are much highlighted in the paper, but it is not needed.
Round 2
Reviewer 1 Report
The authors have significantly improved the quality of the manuscript. The manuscript is now suitable for publication.
Reviewer 2 Report
I recommend the acceptance.
Reviewer 4 Report
This is a revised version of the manuscript that I have reviewed before. I appreciate the efforts of the authors to improve the quality of the article. I am satisfied with responses to my previous review as well as with changes in the manuscript. This allows me to recommend the acceptance of the work.
This manuscript is a resubmission of an earlier submission. The following is a list of the peer review reports and author responses from that submission.
Round 1
Reviewer 1 Report
The manuscript discusses some of the issues with AI algorithms in the Industrial Internet of Things. Try to explain and illustrate the black box problem of artificial intelligence algorithms. The manuscript as a whole makes some contribution to the issues discussed.
Reviewer 2 Report
This paper discusses the algorithm of artificial intelligence for access to the problem of privacy information, a large number of available data allows the use of artificial intelligence algorithms to improve the industrial application in many ways, but as a result of information privacy issues will have negative effect on artificial intelligence, this paper puts forward the relevant example of privacy protection and artificial intelligence solution. At the same time, there are some problems to be improved.
1. This paper does not clarify the practical application value of this topic. Please introduce practical application to prove the feasibility of this project.
2. Maybe introduce some more innovations of this article.
3. An analysis of the interpretability of ai algorithms is not enough.
Reviewer 3 Report
The article presents a timely and interesting research.
The article is well-organized, its structure is logical. The English used is fine. The format of the references is fine as well.
However, since this a review article, the reviewer believes that adding a few more references should be ideal, mainly to the introductory section.
Reviewer 4 Report
The main goal of the paper was not clearly defined in the Introduction. It is not known what the author's contribution to this paper is, and research methodology was not provided.
In the paper are many statements which are not supported with references to literature, for example:
- in the Introduction, rows 35-37: “Having the capability of interpreting data for solving complex problems, AI is also a good fit for factories. It enables 36 industrial systems to process data, perceives their environment, and learn, while building up experience, in order to become better at a task by dealing with it and its data repeatedly.”
- In section 4.1, rows 280-299
In section 4.2 the Authors present and discuss the remaining challenges related to the use of AI in industrial applications: “Bias and fairness” and “Robustness”. Unfortunately, chapter 4.2.1 is not supported by a description of research carried out by the authors or references to literature. On what basis do the Authors identify these new challenges?
The conclusion is too general, and it does not correspond to the purpose of the paper. What was achieved in this work?
Reviewer 5 Report
The paper try to present an overview of privacy preserving and explainable AI in industrial settings.
Unfortunately the authors fail to provide any additional contribution to the already established knowledge of this area of research. The core section, i.e., section 4, lists the well known privacy concerns of explainable AI, but it does not contain any specific recommendation that should be supported by facts, and well-established discussion. Moreover, from method standpoint, authors have failed to identify the type of review, e.g., argumentative, integrative, methodological, systematic, etc. Also, there is not a well formulated motivation for the scope of the work neither what has been the main research question that guided these efforts.
A 'good' review normally challenges previous ideas and contribute to understanding of certain topics, areas or ideas. This means that your review need to go beyond mere description and 'state-of-the-literature' summaries and develop new ideas and ways of thinking.
So overall, i would like to thank the authors for their work and encourage them to continue on this line of research. However, the paper in its current form is not yet ready for acceptance at the journal.